# Damage Mode Analysis of Steel Box Structures Subjected to Internal Blast Loading

Lu-Meng Li [1], Duo Zhang [1,*] and Shu-Jian Yao [2]

1   College of Science, National University of Defense Technology, Changsha 410073, China
2   College of Traffic & Transportation Engineering, Central South University, Changsha 410083, China
*   Correspondence: zhangduo@nudt.edu.cn

**Abstract:** Steel box structures widely exist in vehicles, ships, and buildings, and internal explosions are one of the primary ways to destroy such targets. In this study, a rapid prediction method for the damage degree evaluation of steel box structures subjected to internal blast loads was proposed. First, the main influencing factors were identified through dimensionless analysis. Next, numerical simulations were conducted to further investigate the key influencing factors and different damage modes that were classified according to their characteristics. The non-dimensional $D_{in}^*$ for damage analysis applicable to the internal explosions and the equations describing the deformation of the wall plate were proposed, followed by a comparative study of the damage features of anisotropic box structures with different structural dimensions. The influence of the coupling relationship between structural dimensions and blast loads on the damage modes was analyzed and three competing mechanisms of material failure were studied to analyze and classify the mode of breach expansion. Finally, a large number of experiments were analyzed to verify the analysis method.

**Keywords:** internal blast; non-dimensional; dynamic response; damage modes

## 1. Introduction

The steel box structure is the basic structural unit of a vehicle carriage, ship compartment, or building room, and the multi-cabin structure is the basic internal structural form of vehicles and ships. Different from the external explosion, there are special blast loads, failure modes, and damage distributions when structures are subjected to internal explosions [1]. After the detonation of the explosive, the shock wave is reflected several times within the structure and collects at the edges and corners of the cabin [2]. When the wall plates break, high-pressure gas will be released from the openings to the neighboring cabin layer to cause further damage. The specificity of the internal blast loads also makes the damage mechanism and failure modes of the components more diverse [3–5].

The process of the internal explosion and dynamic response of the structure is very complex and involves a large number of variables, which have not yet found a ready mathematical equation to express. By using dimensional analysis, we can identify the main reasonable parameters and reduce the number of independent variables in theoretical research. In engineering, this can help design more reasonable model experiments to reveal the physical essence of the problem and clarify the main influencing factors [6]. Dimensional analysis has a long history of development in the study of the dynamic response of structures under impulsive loadings, from the dynamic response of members under impulsive loadings, to the deformation of the plate and shell, and the dynamic response of the box chamber structure under internal blast loadings, which is summarized as follows Table 1 by dimensionless numbers.

**Table 1.** Dimensionless numbers of dynamic response.

| Category | Dimensionless Number | Parameters | Scope of Application/Concerns |
|---|---|---|---|
| Dynamic response damage number of the component | $Dn = \frac{\rho V_0^2}{\sigma_0}$ | where $V_0$ is the impact velocity, $\rho$ and $\sigma_0$ are the density and yield strength of the material. | Characterize the dynamic response and damage of material under impact loading, which is the ratio of the impact load to the resistance of the material [7]. |
| | $Dn = \frac{I^2}{\rho \sigma_0 h^2}$ | where $I$ is the unit impulse, $I = \rho h V_0$, $h$ is the thickness of the component | The dynamic response damage number of the structure under blast loading was obtained based on the impulse equivalence principle [8]. |
| | $Rn = \frac{\rho V_0^2}{\sigma_0}\left(\frac{l}{h}\right)^2 = \frac{I^2}{\rho \sigma_0 h^2}\left(\frac{l}{h}\right)^2$ | where $l$ is the length of the component | Considering the effect of structural dimensions [9]. |
| Dimensionless damage number and energy criterion for plate and shell deformation | $\phi_q = \frac{I}{2h^2 (b l \rho \sigma_0)^{1/2}}$ | where $h$ is the plate thickness, $b$ and $l$ are the plate width and length, respectively | The dimensionless damage number describes the deformation of a component such as a plate or a shell [10]. |
| | $\lambda = \frac{\rho V_0^2 L^2}{M_0 h}$ | where $M_0$ is the bending moment, $M_0 = \sigma_0 h^2 / 4$. | Based on the theory of bending and the membrane response of members, the deflections of plates and beams under impact loads are studied [11,12]. |
| | $\nu = \frac{Q_0 L}{2 M_0}$ | where $Q_0$ is the transverse shear force and $M_0$ is the bending moment | Considering the effect of a shear problem [13]. |
| | $\frac{W}{\sigma_0 h L^2} = k$ $\frac{y}{l} = f\left(\frac{W}{\sigma_0 h L^2}\right)$ | where $W$ is the explosive energy, $\sigma, h$ and $l$ are the yield strength, thickness and length of the wall plate, respectively, $y$ is the member deformation, and $k$ is a constant | To study the problem of an explosion forming, it is proposed that the total deformation energy of the plate is proportional to the explosive energy, and the explosive energy utilization is similar for thin plates during explosion forming. A functional expression related to the deformation of the component is proposed [14]. |
| Dimensionless damage number for box structure | $D_{in} = \frac{Q}{\sigma_0 L^2 h}$ | where $Q$ is the explosive energy | Based on the analysis of the dynamic response of the box structure subjected to internal blast loads, a dimensionless damage number describing the degree of damage can be obtained [15]. |
| | $\phi_q = \frac{Q}{\sigma_0 h^2 L}$ | where $Q$ is the explosive energy | The effect of membrane forces on the degree of wall panel deformation is considered [16]. |
| | $D_m = \frac{Qm}{\sigma_0 L h R}$ | where $m$ is the explosive mass, $R$ is the shortest distance between the explosive and the bulkhead. | Damage law of cabin structure under two implosion loading was studied [17]. |
| | $B_c = \frac{W C_s^2 L_t}{\sigma V h}$ | where $L_t$ is the characteristic length of a single box cabin, $C_S$ is the material sound velocity, and $V$ is the volume of the cabin | The maximum deflection and crack length were used to characterize the dimensionless damage number by considering the wall plate deformation and fixed boundary tears [18]. |

The anisotropy of the multi-cabin structure (different materials and/or structural dimensions of the wall plates) will have a more complex coupling effect with the blast loads. Different structural dimensions of the plates, with different resistance and guidance to the blast loads, failure modes, and pressure relief time are also different. Anisotropic cabin structures are more common in daily life, and it is more relevant to study the damage

effects of their internal explosions; however, there are gaps in the related research. In this study, based on the characteristics of the internal blast loads and the mechanism of the structural response, the failure mechanism and damage characteristics of the box structure subjected to internal blast loading were analyzed. Different damage modes are classified according to the characteristics and degree of damage. The non-dimensional $D_{in}^*$ was proposed for damage analysis that was applicable to internal explosions, and the equation describing the deformation of the wall plate was proposed. Based on the study of isotropic multi-cabin structures, a comparative study of the damage features of anisotropic multi-cabin structures with different structural dimensions was conducted. The influence of the coupling relationship between structural dimensions and blast loads on the damage modes of the blast-loaded cabin was analyzed; three competing mechanisms of material failure were studied to analyze and classify the mode of breach expansion. A large number of experiments were analyzed using this method to verify the validity of the method. A foundation was built for the development of damage models for multi-compartment structures affected by internal explosions, which provides a reference that can help engineers design box structures and predict and evaluate the extent of structural damage under internal explosions. A foundation was also built for the development of damage models for box structures subjected to the internal explosions, which provides a reference that can help engineers design box structures and evaluate the structural damage degree under internal explosions.

## 2. Dimensional Analysis

When an explosion occurs inside a box structure, the degree of damage can generally be assessed and measured by the maximum deflection of the wall plate $\delta$. The degree of structural damage is influenced by many factors, such as total explosive mass $W$, explosive blast energy $E$, structural material yield strength $\sigma_y$, material density and sound velocity $\rho_s$ and $C_S$, air density and sound velocity $\rho_0$ and $C_0$, structural characteristic dimension $L$, and wall plate thickness $h$. The main influencing factors and the relationship between them were identified through dimensional analysis. The damage to the box-shaped structure is expressed as

$$\delta = f(W, E, \sigma_y, \rho_s, C_s, \rho_0, C_0, L, H) \tag{1}$$

The explosion is assumed to occur in ideal air, where the speed of sound *late deformation energy ratio*. $C_0$ and density $\rho_0$ are considered to be constant. The problem has eight physical quantities, including seven independent variables and one dependent variable, that involve three basic measures of mass $M$, length $L$, and time $T$, as shown in Table 2.

**Table 2.** Variables and quantitative.

| | Dependent Variable | Independent Variables | | | | | | |
|---|---|---|---|---|---|---|---|---|
| Variable | $\delta$ | $W$ | $E$ | $\sigma_y$ | $\rho_s$ | $C_S$ | $L$ | $H$ |
| Quantitative | $L$ | $M$ | $ML^2T^{-2}$ | $ML^{-1}T^{-2}$ | $ML^{-3}$ | $LT^{-1}$ | $L$ | $L$ |

Using the Vaschy–Buckingham theorem, five dimensionless quantities can be obtained, including four dimensionless independent variables and one dimensionless dependent variable

$$\Pi_0 = F(\Pi_1, \Pi_2, \Pi_3, \Pi_4) \tag{2}$$

where

$$\Pi_0 = \frac{\delta}{h}, \Pi_1 = \frac{E}{\sigma_y L^2 h}, \ \Pi_2 = \frac{W}{\rho_s L^2 h}, \Pi_3 = \frac{\rho_s C_s^2}{\sigma}, \Pi_4 = \frac{L}{h} \tag{3}$$

to obtain the dimensionless function

$$\frac{\delta}{h} = F\left(\frac{E}{\sigma_y L^2 h}, \frac{W}{\rho_s L^2 h}, \frac{\rho_s C_s^2}{\sigma}, \frac{L}{h}\right) \tag{4}$$

The above equation shows that the structural deformation deflection is a function of the four dimensionless expressions above, and each expression has a clear physical meaning: $\frac{E}{\sigma_y \varepsilon L^2 h}$ is the ratio of blast energy to structural deformation energy, which characterizes the transformation and consumption between blast energy and plastic deformation energy of the wall plate; $\frac{W}{\rho_s L^2 h}$ is the ratio of explosive mass to the mass of the member, which characterizes the transformation and consumption between the explosive energy and the kinetic energy of the wall plate; $\frac{\rho_s C_s^2}{\sigma}$ for the ratio of material strength and stiffness, which characterizes the ratio of the energy generated by rupture or tearing to the energy deformed, and $\frac{L}{h}$ for the geometric scale of the structural elements. The energy generated by the explosion is mainly divided into wall kinetic energy, plastic deformation energy, and rupture fracture energy, and a portion of the energy is transferred to the adjacent box through the breakage [19]. Compared to other energy, the fracture energy consumed by the wall panel rupture is very small. A simplified equation, expressed in the form of an exponential product, is shown in Equation (5)

$$\frac{\delta}{h} = \left(\frac{E}{\sigma_y L^2 h}\right)^{\beta} \left(\frac{W}{\rho_s L^2 h}\right)^{\gamma} \left(\frac{L}{h}\right)^{\theta} \tag{5}$$

For the box structure subjected to the internal blast loading, the dimensionless number $D_{in}^*$ is proposed as

$$D_{in}^* = \frac{\sqrt{EW}}{hab\sqrt{\rho\sigma_y}} = \frac{\sqrt{E}}{\sqrt{\sigma_y hab}} \frac{\sqrt{W}}{\sqrt{\rho hab}} \tag{6}$$

where $E$ is the total energy of the explosion, $W$ is the explosive equivalent, $a$, $b$, and $h$ are the length, width, and thickness of the wall plate, respectively.

## 3. Numerical Simulation Methods and Validation

### 3.1. Finite Element Model

The scaled-down experiments on the cabin structures subjected to internal explosions were performed by Yao [6]. The test specimens are square-shaped single cabin structures with extended boundaries, with lengths of 60 cm, 45 cm, and 30 cm, thicknesses of 4 mm, 3 mm, and 2 mm, respectively, and a round hole in the middle of the top plate for placing explosives in the center of the cabin. In numerical simulations, using the finite element software LS-DYNA, the finite element models were established, as shown in Figure 1. The structure was meshed using the SHELL163 element type, while the explosive and air were meshed using the SOLID164 element type. The mesh size of the structure and air is 4 cm. The explosive is filled in the centre of the air using a fluid-solid coupling algorithm. The surface of the air is set up with no-reflection boundary conditions, as in Figure 1.

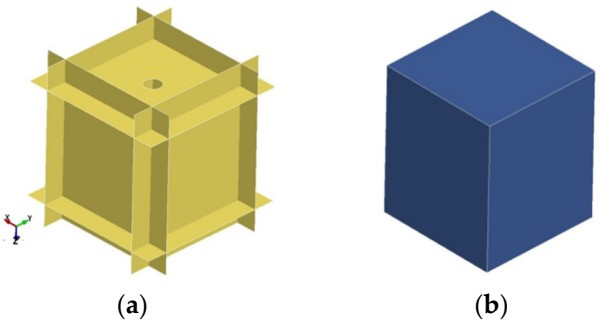

(**a**)　　　　　　　　　(**b**)

**Figure 1.** Calculation model [6]. (**a**) Box structure and coordinate; (**b**) Air domain.

### 3.2. Material Models and Parameters

The air is defined using the keyword *MAT_NULL, which is modeled by a linear polynomial equation of state (EOS_LINEAR_POLYNOMIAL) [20]. The pressure can be expressed as:

$$P = C_0 + C_1\mu + C_2\mu^2 + C_3\mu^3 + \left(C_4 + C_5\mu + C_6\mu^2\right) \tag{7}$$

where $P$ is the overpressure; $E$ is the internal energy per unit volume; $\mu = \rho/\rho_0 - 1$, where $\rho_0$ is the initial density and $\rho$ is a reference density; $c_1 = c_2 = c_3 = c_6 = 0$ and $c_4 = c_5 = 0.4$. The internal energy $E$ of air is 0.25 MPa and the density $\rho$ is 1.29 kg/m$^3$. The keyword *MAT_HIGH_EXPLOSIVE_BURN is used for explosives, and the JWL equation of state is used to model the pressure generated by the explosion:

$$P = A\left(1 - \frac{\omega}{R_1 V}\right)e^{-R_1 V} + B\left(1 - \frac{\omega}{R_2 V}\right)e^{-R_2 V} + \frac{\omega E}{V} \tag{8}$$

where $P$ is the pressure of high explosive, $V = v/v_0$ is the relative volume, $v$ is the specific volume, $v_0$ is the initial specific volume, and $E$ is the internal energy per unit volume. The parameters for TNT and *EOS_JWL are shown in Table 3.

**Table 3.** Material model of TNT [6].

| $P$ (kg·m$^{-3}$) | $VCJ$ (m·s$^{-1}$) | $PCJ$ (GPa) | A (Gpa) | B (Gpa) | $R_1$ | $R_2$ | $\omega$ |
|---|---|---|---|---|---|---|---|
| 1601 | 6850 | 21 | 373.77 | 3.231 | 4.15 | 0.95 | 0.35 |

The test devices are welded by Q235B steel plates, and the yield stress and failure strain of the steel box is calculated using the MAT_PLASTIC_KINEMATIC [20] model. There are differences in mechanical properties due to different thicknesses of steel plates. The parameter settings referred to research by Yao [6] and Chen [21], as shown in Table 4.

**Table 4.** Mechanical properties of Q235B steel plate.

| $h$ (mm) | $\rho$ (kg·m$^{-3}$) | $v$ | $\sigma_0$ (MPa) | $Et$ (MPa) | $E$ (GPa) | $F_s$ |
|---|---|---|---|---|---|---|
| 2 | | | 370 | 485 | | 0.30 |
| 3 | 7800 | 0.3 | 368 | 484 | 210 | 0.29 |
| 4 | | | 360 | 480 | | 0.38 |

### 3.3. Validation Results

Measure the central deflection of the front, back, left, and right wall plates and take the average as $\delta$. The deflection of the wall plates were dimensionless. In the equation $\delta = \omega/h$, $\omega$ is the deflection and $h$ is the thickness. The experimental results are compared with the numerical results and listed in Table 5; the numerical simulation results are within 3% error, which is closer to the measured results.

**Table 5.** Experimental and numerical simulation results [6].

| $L$ (mm) | $h$ (mm) | $W$ (g) | Measured Deflection Ratio | Simulated Deflection Ratio | Error |
|---|---|---|---|---|---|
| 300 | 2 | 23.9 | 11.95 | 12.25 | 2.5% |
| 450 | 3 | 84.3 | 13.13 | 13.3 | 1.3% |
| 600 | 4 | 194.69 | 13.33 | 13.6 | 2.0% |

The experimental results of deflection and dynamic deformation processes were compared with numerical simulation results, and the results are in good agreement. It was proven that the numerical simulation method can effectively simulate the deformation characteristics of the box structure under internal blast loadings with high reliability.

### 3.4. Calculation Model of Multi-Box Structures

According to the study by Zhang [22], the area of the ship's compartment is generally around 10 m², the height between decks is generally 2.4~2.8 m, and the surrounding wall thickness is about 6 mm. According to the geometric similarity rule, $3 \times 3 \times 3$ multi-cabin structures with extended boundaries were designed, each with a length–width –height dimension of 120 cm and a wall thickness of 3 mm. A comparative study of the damage characteristics under the implosion of multi-cabin structures with different structural dimensions was conducted. Since the quasi-static pressure is related to the volume of the structure [23], the kinetic energy of the wall plate subjected to the blast is related to its mass, and the mass of the wall plates is mainly determined by the area when the thickness of the wall plate material is the same. Therefore, four groups of $3 \times 3 \times 3$ multi-box structures with the same volume and chamber wall aspect ratios $\gamma$ of 1, 2, 3, and 4 were designed. One set of walls has an equal area and aspect ratio $\gamma$. The other two sets of walls have equal side lengths of 120 cm on one side and a $= 120 \times \sqrt{\gamma}$ cm and b $= 120/\sqrt{\gamma}$ cm on the other two sides. The whole chamber structure has a length to width ratio $\gamma$ and length to height ratio $\sqrt{\gamma}$ with the dimensions shown in Table 6.

Using the numerical simulation method described in the previous section, the finite element model is shown in Figure 2. As shown in Figure 3, the A-profile, B-profile, and C-profile are made along parallel x-y, x-z, and y-z axes, respectively, with the center of the blast-loaded cabin. The length, width, and height of the single cabin are *a*, *b*, and 120 cm, respectively.

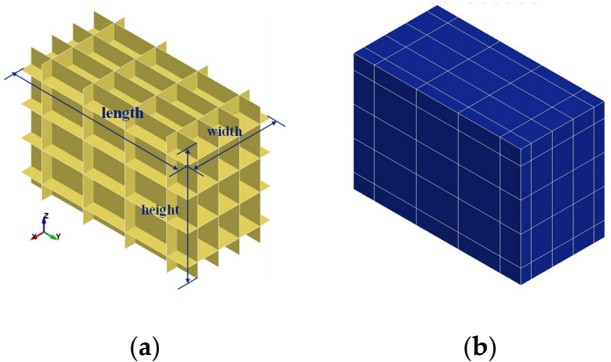

**Figure 2.** Finite element model. (**a**) Box structure and coordinate system; (**b**) Air domain.

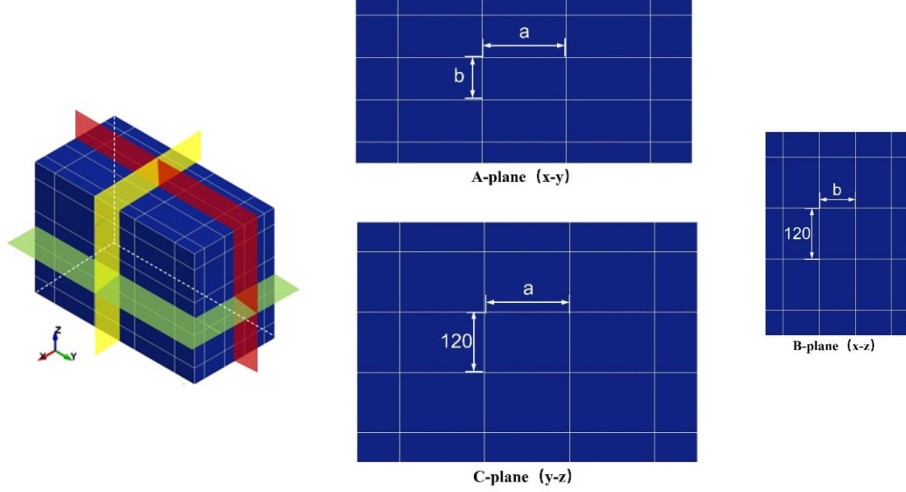

**Figure 3.** Multi-box structural section and structural dimensions (Unit: cm).

**Table 6.** Characteristic dimensions of multi-box structures.

| No. | $\gamma$ | Dimensions of the Single Box/cm | $h$ (mm) | Amount of Shell Units | Amount of Air Units |
|---|---|---|---|---|---|
| E-I | 1 | 120 × 120 × 120 | 3 | 172,800 | 1,728,000 |
| E-II | 2 | 84.8 × 169.7 × 120 | 3 | 185,344 | 1,734,000 |
| E-III | 3 | 69.3 × 207.8 × 120 | 3 | 194,304 | 1,747,200 |
| E-IV | 4 | 60 × 240 × 120 | 3 | 201,600 | 1,728,000 |

## 4. Analysis of Calculation Results

### 4.1. Different Failure Modes of Isotropic Box Structure Subjected to the Internal Blast Loadings

Nurick and Shave [24] classified the failure modes of square plates under impact loadings into large inelastic deformation, tensile tear damage, and shear damage. By analyzing the damage characteristics of typical box structures under different TNT equivalents, the failure modes were subdivided concerning the division by Yao [25]. Figure 4a shows the shock wave impact on the wall of the box structure. When the strain was greater than the plastic strain and less than the fracture strain of the materials, the wall plate presented mode I plastic large deformation. According to the first break location, tensile tear damage could be subdivided into the following damage modes: mode II center breach of the plates. The plastic hinge lines were produced on the two diagonals when impacted by the shock wave and gradually approached the center. When the strain in the central region was larger than the critical value, the wall plate produced a central breach, as shown in Figure 4b. In the relatively closed cabin structure, the shock wave would converge in the corner and edge, and the cabin structure was prone to form a breach and expand at the corner or show edge tearing (mode III), as shown in Figure 4c. When the explosion equivalent was larger, the transverse shear strain of the plate reached the limit first. The damage mode transformed from tensile damage to mode IV shear damage, and the boundary was approximately straight shear damage, as shown in Figure 4d.

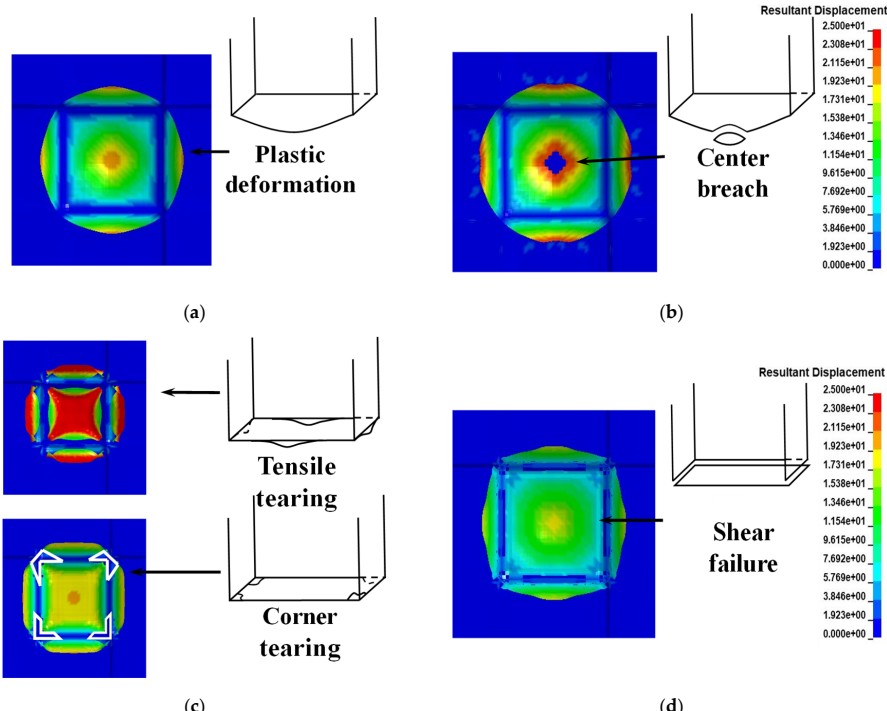

**Figure 4.** Failure modes of the blast-loaded cabin: (**a**) Mode I Plastic deformation ($W$ = 3 kg, $t$ = 2.50 × $10^{-3}$s); (**b**) Mode II Center breach ($W$ = 6.25 kg, $t$ = 4.65 × $10^{-3}$s); (**c**) Mode III Tensile tearing ($W$ = 11 kg, $t$ = 1.05 × $10^{-3}$s, $W$ = 15 kg, $t$ = 1.35 × $10^{-3}$s); (**d**) Mode IV Shear failure ($W$ = 55 kg, $t$ = 0.45 × $10^{-3}$s).

### 4.2. Effect of Structural Dimensions on the Transition from Damage Mode I to Mode II

Figure 5 shows the damage results of the E-I, E-II, and E-IV test models subjected to the internal explosion with a TNT equivalent $W$ = 6.25 kg. Points a, b, and c are the center unit of the A-, B-, and C-plane plates of the model structure. Point d is the cell in the B-plane (x-z direction) of the E-II model, approximately 15 cm from point b along the positive direction of the z-axis. After the explosion, the pressure pulse impacted the plates of the blast-loaded cabin in the E-I model. The plastic hinge lines were produced on the two diagonals and gradually approached the center. At this stage, the central unit accelerates, the deflection becomes greater and a plastic hinge ring is formed. When the pressure pulse reaches the static plastic collapse pressure, the wall plate is not further damaged.The central units of the wall plates were no longer accelerated, and the plastic deformation consumed the kinetic energy of the system. When the deformation in the region exceeded the fracture strain, the center of the wall plates produced a breach, as shown in Figure 6a. Comparing the damage feature of the anisotropic multi-cabin structures with the same volume and different structural dimensions, the E-II model had a larger C-plane in the blast-loaded cabin, a smaller explosion distance, and larger kinetic energy that was consumed in the plastic deformation process. Until the velocity of unit deceleration is nearly 0, the strain in the central region does not exceed the fracture strain, the C-plane plates had not observed a breach in the center, and deformation slowly converged to a normal value. The units of the B-plane plate accelerated under the pressure impulse and plastic deformation consumed kinetic energy. In the process of unit deceleration, at $t$ = 1.8 ms, the units of the B-plane plate had not decelerated to 0, it was subjected to the second impulsive loadings, and the wall plates were further damaged after the original deformation. Near the plastic hinge ring d point at the weaker stress concentration, which was the first to be destroyed to form the pressure relief port, energy was consumed and released without causing further damage to other wall plates, as shown in Figure 6b. Compared with the E-I and E-II test models, the C-plane plate of the blast-loaded cabin in the E-IV model was much larger and the explosion distance was smaller. The process of generating plastic hinges and gathering them to the center of the plates consumed more energy. The remaining energy did not make the other wall plates produce a significant break, and the large plastic deformation were observed on the walls, as shown in Figure 6c.

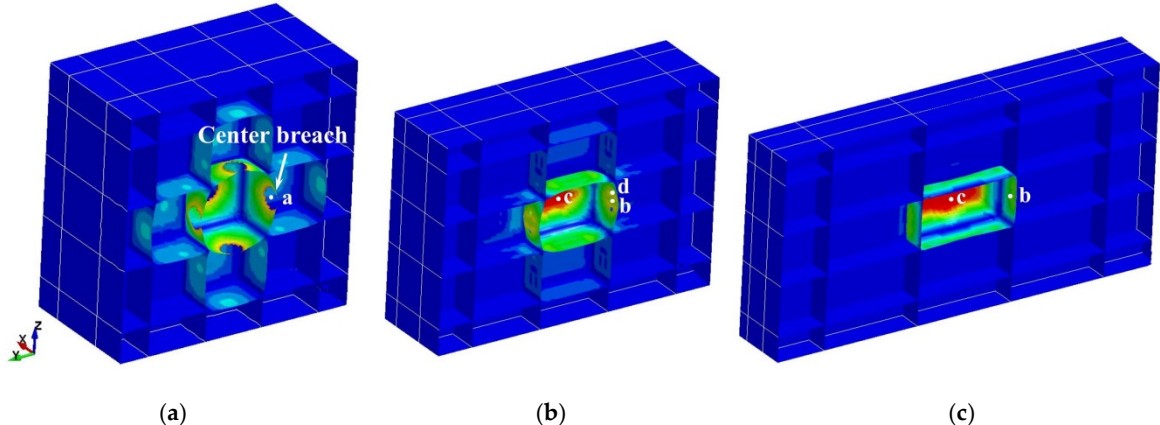

**Figure 5.** Damage of multi-box structures of different structural dimensions (Unit: cm): (**a**) 120 × 120 × 120; (**b**) 84.8 × 169.7 × 120; (**c**) 60 × 240 × 120.

The analysis above indicates that the transformation of the wall plates from damage mode I to mode II under internal blast loadings is mainly the transformation and consumption between explosive energy, the kinetic energy of the wall plate units, and plastic deformation energy. The strain energy of a plate is expressed as $\sigma \varepsilon h a b$ in Zheng's study [14], where $\sigma$ is yield stress, $\varepsilon$ is the average strain, $h$ is the thickness, and $a$ and $b$ are the length and width, respectively. In the non-dimensional number $D^*_{in}$, $\frac{E}{hab\sigma_y}$ is the ratio of explosive

energy to the deformation energy, and the transformation of damage occurs when a wall plate with different areas exceeds the limit strain under internal blast loadings.

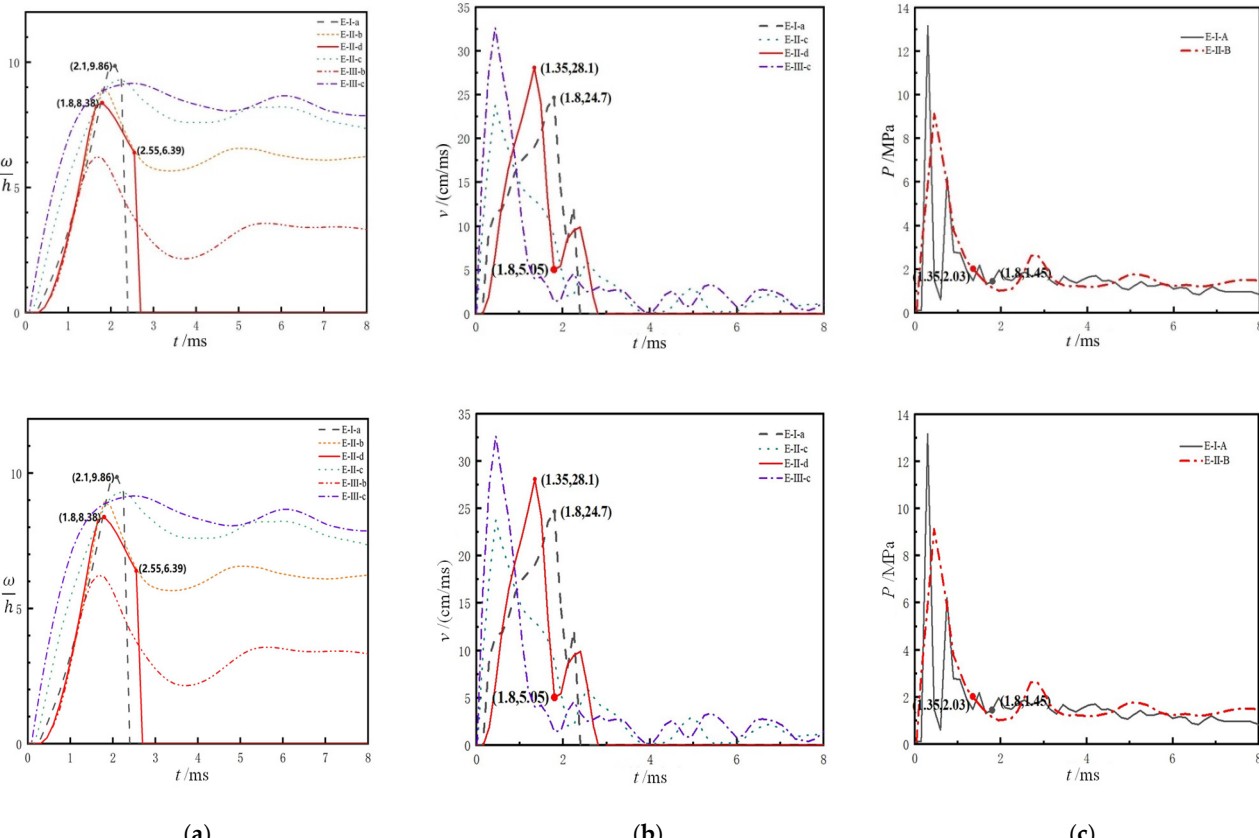

**Figure 6.** The variation of deflection, velocity, and pressure at different marker points. (**a**) Variation of deflection; (**b**) Variation of velocity; (**c**) Variation of pressure.

The aspect ratio will also influence how the damage mode transforms the same area with different wall plate aspect ratios, as demonstrated in Table 7 for the same internal blast equivalent damage effect for the same area with different wall plate aspect ratios.

**Table 7.** The damage features of the same area & different aspect ratios of the wall plates.

| Explosion Equivalent W (kg) | Wall Dimensions $a \times b$ (m) | $\gamma$ | Wall Plate Area (m²) | Non-Dimensional $D_{in}^*$ | Dimensionless Deflection ($\omega \cdot h^{-1}$) | Damage Mode |
|---|---|---|---|---|---|---|
| 6.25 | 1.2 × 1.2 | 1 | 1.41 | 0.872 | 10.33 | Center breach |
| 6.25 | 1.697 × 0.848 | 2 | 1.41 | 0.872 | 6.66 | Plastic large deformation |
| 6.25 | 2.4 × 0.6 | 4 | 1.41 | 0.872 | 3.4 | Plastic large deformation |

The smaller the aspect ratios (the closer to the square plate), the larger the center deflection of the wall plates is for the same area subjected to internal blast loadings. The damage mode of the wall plates changed from plastic large deformation into a central breach when the distortion exceeded the fracture strain.

### 4.3. Effect of Explosive Chamber Structural Dimensions on the Transition from Damage Mode III to Mode IV

The damaged C-planes (y-z direction plane) for the E-I and E-III models are shown in Figure 7. The effective plastic deformation curves of the model points are shown in Figure 7c. When the E-I model was subjected to a TNT equivalent W = 25 kg implosion,

the central deformation of the plates increased rapidly. At t = 0.75 ms, there was almost simultaneous tearing of the bulkhead plate edges and detachment from the structure, while flap fractures were produced in the box structure's corners. Comparing the damage of the E-III test model, the C-plane with the largest area and smallest blast distance is the first to form a central breach (mode II), with an effective plastic strain of 0.268. The A-plane experiences tearing that starts in the middle of the long edge and spreads to the corners, but the central portion of the plate maintains a relatively flat shape during failure. Deformation is primarily concentrated near the boundary with an effective plastic strain of 0.0896, which shows a tensile tear at the edge of damage mode III. The B-plane has a smaller area, a lower mass, and a higher beginning velocity. The damage is more restricted to the support, the wall plate center deformation is also the smallest, and the effective plastic strain is 0.033, which shows shear damage (mode IV).

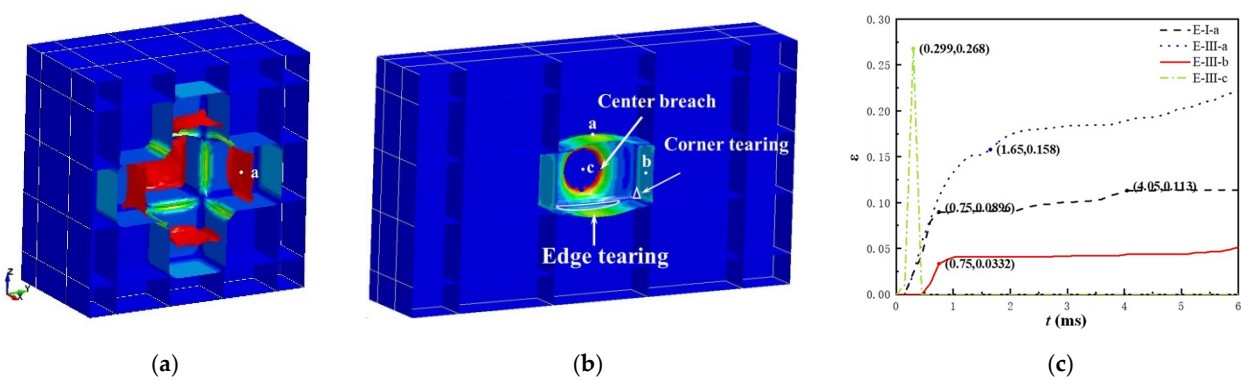

(a) (b) (c)

**Figure 7.** Damage results of multi-box structures with different structural dimensions. (**a**) E-I model t = 1.80 ms; (**b**) E-III model t = 0.90 ms; (**c**) Variation of effective plastic strain.

　　　Plastic deformation of the rectangular plates is observed under strong impact loadings, mainly bending that yields at the midpoint of the long side, tensile yielding in the center of the plate, and easy shear yielding for a smaller mass area of the wall plates. Jones [26] proposed that the critical velocity for the transformation of damage mode III to mode IV depends only on the material properties and corresponds to the $\frac{W}{hab\rho}$ part of the modified non-dimensional, where the ratio is the explosive charge to the mass of the wall plates. Under the same implosion loads, the same material wall plate, with a smaller area, is easier to reach the critical velocity and easier to occur shear damage.

　　　As shown in Figure 7b, the damage to the different wall panels in the E-III test chamber is different and distinguishes the first damage location, which can be divided into the center of the wall panel, the midpoint of the edge, and the corner. Under imposed load and inertia force, the fracture expands in different ways due to different energy consumption and different final fracture sizes. In reference to the crack classification mode, according to the relationship between the stress and the direction of the crack expansion surface, the extension mode of the break can be divided into three categories. (a) Open type (Type I) crack: positive stress is perpendicular to the crack face, and the direction of expansion is perpendicular to the stress. (b) Slip-open (Type II) crack: the force is along the direction of the crack face, and the shear stress and the crack expansion direction are parallel (c) Tear-open (Type III) crack: the shear stress and the crack expansion direction are perpendicular, and the crack face slides relatively [27,28]. These openings are shown in Figure 8.

　　　The location of the openings in the different walls and the special points on the expansion path of the openings are shown in Figure 9. The first damage location point a, point c, and point e are subject to less stress, and fracture deformation is small for the slip openings (Type II), external load along the direction of the opening, and shear damage. After the damage is subjected to the blast loading, the breach will be extended from point a, c, and e along the vertical load direction to point b, c, and e, respectively. The extension

process belongs to the tearing type (type III) and tensile damage occurs. According to the various methods of expanding the opening, the energy used is different, and the size of the final opening is different.

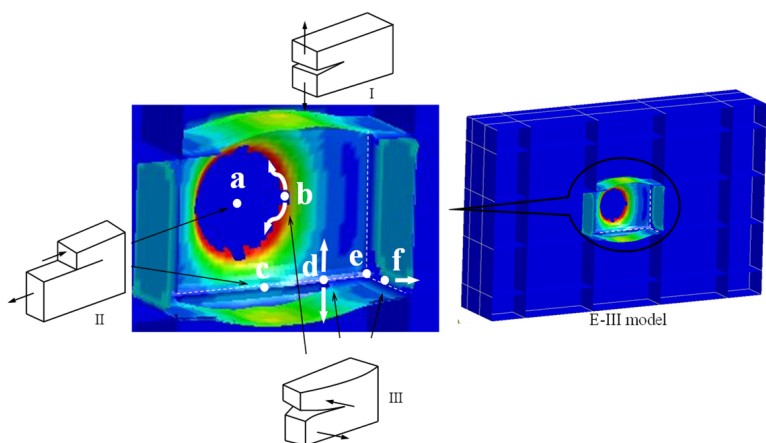

**Figure 8.** Type of openings.

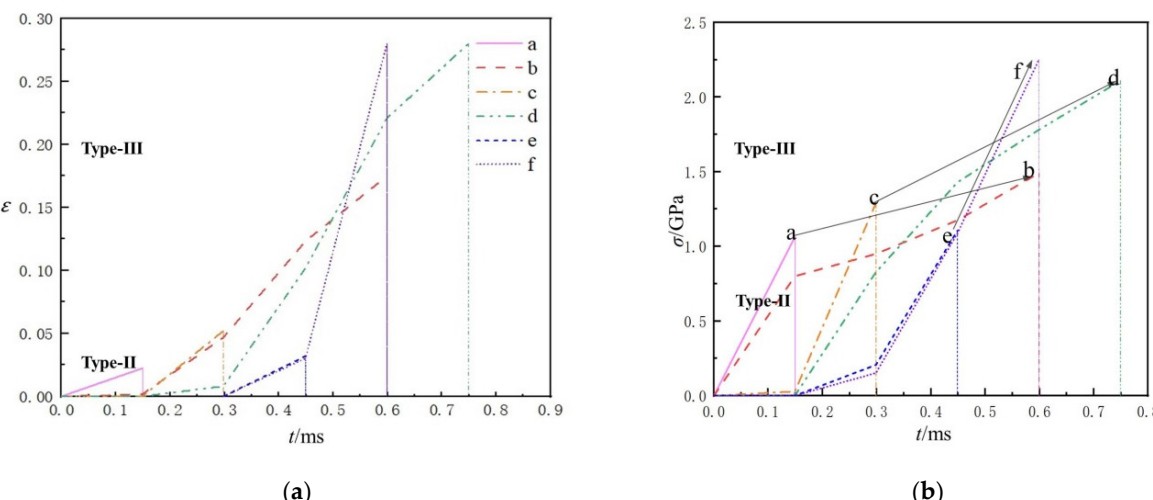

(**a**)                    (**b**)

**Figure 9.** Stress and plastic strain curves at the marker pilot. (**a**) Variation of effective plastic strain; (**b**) Variation of effective strain.

### 4.4. Damage Degree Prediction Using Dimensionless Number Din*

Extensive numerical simulations were carried out to obtain the damage degree of the box chamber structure. The relationship between the non-dimensional deflection and the non-dimensional number $D_{in}^*$, as well as the division of the damage modes, are shown in Figure 10. Based on the failure mechanisms of the box chamber structure, the equation describing the deformation of the wall plate is proposed based on the dimensionless number $D_{in}^*$

$$f(\omega^*) = \left(\frac{W}{\rho hab}\right)^{\beta} \left(\frac{E}{hab\sigma_y}\right)^{\gamma} \left(\frac{a}{h}\right) \tag{9}$$

where $\omega^*$ is the dimensionless deflection (ratio of wall plate deflection to thickness), $\frac{E}{hab\sigma_y}$ is the ratio of the explosive energy to the deformation energy of the wall plate, $\frac{W}{\rho hab}$ is the ratio of the explosive equivalent to the mass of the wall plate, and $\frac{a}{h}$ is the structural scale effect. The influence factor $\alpha$ is proposed, which is related to the relative strength of impact, and $\alpha_1 = 1 - e^{-\frac{E}{\sigma_y hab}}$ is the yield strength influence factor when $\alpha_1 = 1$ is completely dominated

by yield strength; $\alpha_2 = 1 - e^{-\frac{W}{\rho hab}}$ is the inertia influence factor when $\alpha_2 = 1$ is completely dominated by inertia.

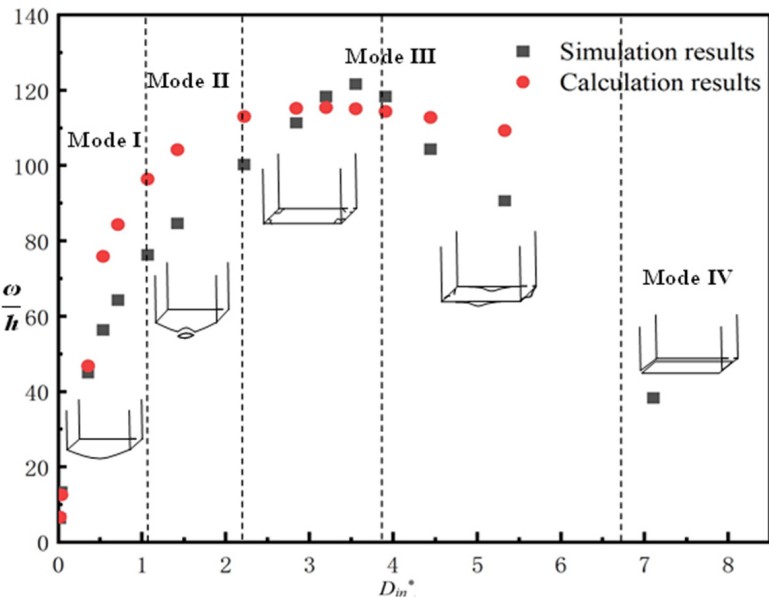

**Figure 10.** The variation of deflection and damage modes of box structures with $D_{in}^*$.

When $D_{in}^* < 0.37$ the explosion equivalent is small, the wall plate subjected to the impact force will be flattened or become bent, and all physical phenomena will occur following the easiest route. When the impact load is small, the wall plate deformation by inertia is relatively large and more easily flattened. With the increase of the equivalent, the impact of yield strength increases, the wall plate is more likely to show bend deformation, and the wall plate presents mode I large deflection plastic deformation. At this time, the wall plate deformation degree formula is

$$f(\omega^*) = \left(\frac{W}{\rho hab}\right)^{\alpha_1 + \alpha_2} \left(\frac{E}{hab\sigma_y}\right)^{2-\alpha_1-\alpha_2} \left(\frac{a}{h}\right) \tag{10}$$

when $0.37 < D_{in}^* < 1.15$, the wall plate presents mode I large deflection plastic deformation. When $1.15 < D_{in}^* < 2.2$ presents mode II wall plate central breach, the damage degree of the wall plate at this stage is dominated by the yield strength. The degree of deformation of the wall plates, and whether they fracture, depends on the plastic strain and the fracture strain. Whether the wall plate is deformed enough to fracture, as well as the degree of the deformation, depends on the plastic strain and fracture strain.

When $2.2 < D_{in}^* < 6.75$, the implosion load is gathered in the box edge and corner, the wall plate presents mode III corner tensile tear damage, and wall plate deflection with an increase of the dimensionless number first increases and then decreases. When the the edges of a wall plates show tensile tear damage, the deformation of the edge is much larger than the center of the plate, resulting in the center part being "pulled into" the center of the deflection of a small decline. This stage is dominated by yield strength and inertia together. The box structure under the implosion load produces corner rupture or prismatic tearing, the formation of a pressure relief port, and partial energy release.

When $D_{in}^* > 6.75$, the wall plate mode IV shear damage, the degree of damage to the wall plate at this stage, is dominated by inertia. When the wall plate is subjected to internal

blast loading in a very short time at a very high rate from the structure, the center deflection is small and tends to stabilize. At this time, the wall deformation degree formula is

$$f(\omega^*) = \left(\frac{W}{\rho hab}\right)^{\frac{\alpha_1+\alpha_2}{2}} \left(\frac{E}{hab\sigma_y}\right)^{\frac{1-\alpha_1-\alpha_2}{2}} \left(\frac{a}{h}\right) \tag{11}$$

According to the numerical simulation results, the deformation of the wall plate is measured and the damage modes are classified. The measured results are shown in the black square dots in Figure 10. The test condition parameters are substituted into Equations (10) and (11) to calculate the degree of wall plate deformation, which is shown in Figure 10 as red dots. For the same trend of change, the error is within 20%.

To describe the opening area of the blast-loaded cabin, the non-dimensional $\psi$ was introduced as the ratio of the opening area to the wall plate area. Figure 11 showed the variation of the opening of wall plates at different $D_{in}^*$. When $D_{in}^* = 1.15$, a breach was produced in the center of the wall plate and the damage mode was transformed from plastic large deformation to a central breach. With the increase of $D_{in}^*$ the opening area, $\psi$ increases gradually and the wall plates are completely detached from the structure at $D_{in}^* > 3.93$; the detachment time is gradually advanced with the increase of the dimensionless number $D_{in}^*$. When $D_{in}^* = 6.9$, the impulse reached its critical value (i.e., that causing shear failure over the entire plate), the wall plates completely detached from the structure in a short time, and the non-dimensional $\psi$ reached 1.

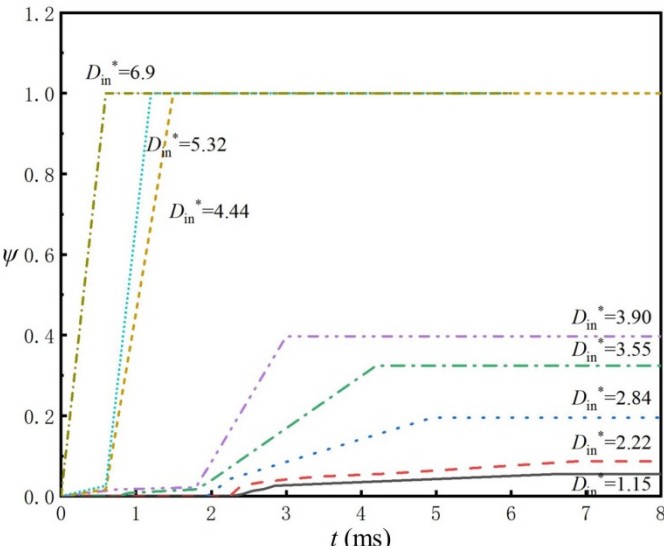

**Figure 11.** The variation in opening.

To have a better understanding of the damage characteristics, a comparative study of the damage effects of box structures with different structural dimensions was conducted based on isotropic box structures.

The anisotropic box structure has different dimensions and areas of wall plates in different directions. Under the condition that the width of the wall plate is certain, the aspect ratio is changed by altering the length $a$, $\beta = a/120$, in which the wall plate material and thickness are the same. As shown in Figure 12, with the consequent variation of the dimensionless number $D_{in}^*$, the trend of central deflection of the plate is similar to that of the isotropic box structure. During the mode I plastic large deformation, mode II central breakage and mode III corner tearing stages, the central deflection of the wall panel increases with $D_{in}^*$ and the peak central deflection increases with $\beta$. The dimensionless deflection of the wall panel decreases rapidly after reaching the peak. During the Mode IV shear damage stage, the dimensionless deflection of the wall panel stabilises at around 20.

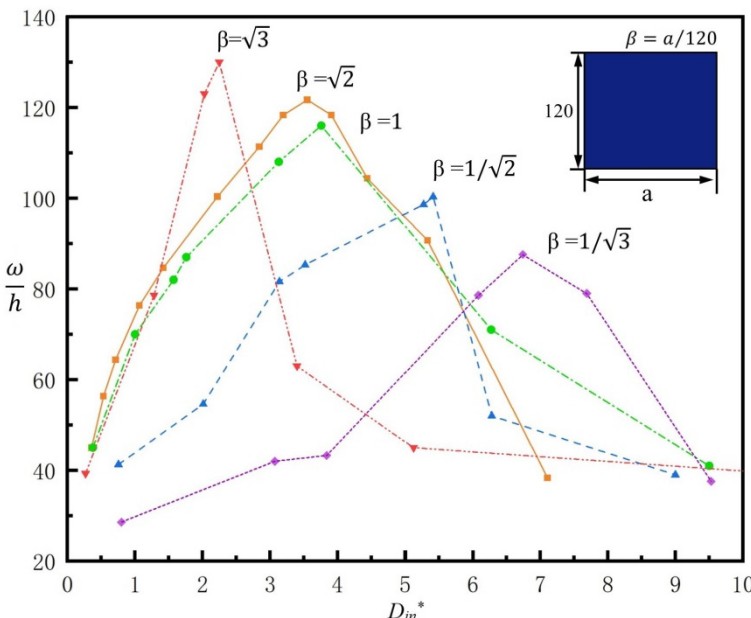

**Figure 12.** The variation of deflection of anisotropic box structure with $D_{in}^*$.

*4.5. Application and Validation of Analytical Methods*

Five different sets of existing experiments were analyzed and verified by the analytical methods detailed above.

Experiment 1: In reference to the internal explosion shrinkage ratio experiments carried out by Yao [6], the test device is a square-shaped single box structure with extended boundaries. The square box structure had side lengths of 300 mm, 450 mm and 600 mm, and plate thickness of 2 mm, 3 mm, and 4 mm, respectively. The top plate in the middle of the circular hole was used to place the TNT explosive in the center of the box.

Experiment 2: A series of experiments on the deformation response of box structures subjected to internal explosions were carried out in reference [29] with a box structure size of 200 mm × 200 mm × 200 mm the plate thickness of 3 mm, 4 mm, and 5 mm, respectively. The side plate is welded to the bottom plate by fillet welding and bolted to the top plate. The PE4 explosive is placed in the center of the box, and the burst distance for each plate is 100 mm.

Experiment 3: Reference [16] carried out a series of internal explosion experiments. A rectangular box structure (explosive cylinder) using clamps of different thickness with dimensions of 800 mm × 800 mm sheet metal specimens were assembled with 36 M16 bolts of steel plate on the specimen clamping restraint, which will have different yields of TNT cylindrical charge that are placed in the center of the box and detonated.

Experiment 4: The size of the 200 mm × 200 mm Q235 metal plate is sandwiched between two 20mm-thick steel plates, and the measurement of its deformation under different equivalent explosive loads is observed [24].

Experiment 5: The experimental setup [30,31] is placed in a water tank of 15 m × 12 m × 10 m. The exposed area of the metal plate of HS steel and mild steel of two materials are measured at 300 mm × 250 mm with a thickness of 2 mm. The PEK I explosives of different equivalents are placed in the channel section of the box chamber device.

The experimental parameters are substituted into Equation (6) to obtain the dimensionless number $D_{in}^*$, and when $D_{in}^* < 0.37$, the dimensionless deflection of the center deformation of each wall is calculated using Equation (10); when $D_{in}^* > 0.37$, it is calculated using equation (11). The material parameters, experimental working conditions, and calculation results and errors of the control experiment are shown in Table 8.

**Table 8.** Experimental conditions and analysis results.

| No. | $W$(g) | Explosive Type | $a \times b$ (mm) | $H$ (mm) | $\sigma$ (MPa) | $D_{in}^*$ | Calculated Result $\omega/h$ | Experimental Result $\omega/h$ | Error |
|---|---|---|---|---|---|---|---|---|---|
| 1 [6] | 12.8 | TNT | 300 × 300 | 2 | 370 | 0.084 | 7.84 | 7.44 | 0.05 |
| | 23.9 | TNT | 300 × 300 | 2 | 370 | 0.158 | 9.66 | 11.95 | −0.19 |
| | 40.2 | TNT | 300 × 300 | 2 | 370 | 0.266 | 13.61 | 17.19 | −0.20 |
| | 43 | TNT | 450 × 450 | 3 | 368 | 0.084 | 7.82 | 7.7 | 0.03 |
| | 84.5 | TNT | 450 × 450 | 3 | 368 | 0.166 | 9.83 | 13.13 | −0.26 |
| | 146 | TNT | 450 × 450 | 3 | 368 | 0.290 | 14.74 | 18.5 | −0.20 |
| | 99 | TNT | 600 × 600 | 4 | 360 | 0.082 | 7.73 | 8.23 | −0.05 |
| | 200.1 | TNT | 600 × 600 | 4 | 360 | 0.168 | 9.77 | 13.33 | −0.26 |
| | 345 | TNT | 600 × 600 | 4 | 360 | 0.290 | 14.69 | 19.5 | 0.24 |
| 2 [29] | 20 | PE4 | 200 × 200 | 3.4 | 233 | 0.280 | 4.55 | 4.79 | −0.050 |
| | 30 | PE4 | 200 × 200 | 3.4 | 233 | 0.421 | 9.20 | 6.85 | 0.25 |
| | 40 | PE4 | 200 × 200 | 3.4 | 233 | 0.561 | 10.32 | 8.08 | 0.21 |
| | 50 | PE4 | 200 × 200 | 3.3 | 233 | 0.723 | 11.73 | 10.48 | 0.10 |
| | 60 | PE4 | 200 × 200 | 3.4 | 233 | 0.842 | 12.03 | 11.70 | 0.0 |
| | 70 | PE4 | 200 × 200 | 3.4 | 233 | 0.983 | 12.69 | 10.74 | 0.15 |
| | 20 | PE4 | 200 × 200 | 4 | 222 | 0.244 | 3.12 | 3.12 | 0.00 |
| | 30 | PE4 | 200 × 200 | 4.1 | 222 | 0.358 | 5.63 | 4.68 | 0.20 |
| | 40 | PE4 | 200 × 200 | 4.1 | 222 | 0.477 | 7.93 | 6.4 | 0.19 |
| | 50 | PE4 | 200 × 200 | 4 | 222 | 0.611 | 8.97 | 7.97 | 0.11 |
| | 60 | PE4 | 200 × 200 | 4 | 222 | 0.734 | 9.63 | 9.25 | 0.03 |
| | 70 | PE4 | 200 × 200 | 4 | 222 | 0.856 | 10.18 | 9.78 | 0.04 |
| | 20 | PE4 | 200 × 200 | 5.1 | 263 | 0.176 | 1.77 | 1.82 | −0.02 |
| | 30 | PE4 | 200 × 200 | 5.1 | 263 | 0.264 | 2.82 | 2.72 | 0.03 |
| | 40 | PE4 | 200 × 200 | 5.1 | 263 | 0.352 | 4.43 | 3.62 | 0.22 |
| 3 [16] | 55 | TNT | 800 × 800 | 1.8 | 360 | 0.058 | 19.50 | 24.27 | −0.19 |
| | 110 | TNT | 800 × 800 | 3.7 | 320 | 0.060 | 9.43 | 9.16 | 0.03 |
| | 110 | TNT | 800 × 800 | 2.3 | 317 | 0.097 | 17.66 | 23.21 | −0.23 |
| | 110 | TNT | 800 × 800 | 2.7 | 322 | 0.082 | 14.48 | 18.03 | −0.19 |
| | 200 | TNT | 800 × 800 | 4.8 | 317 | 0.084 | 8.16 | 7.87 | 0.03 |
| 4 [24] | 9.1 | TNT | 200 × 200 | 1.6 | 237 | 0.213 | 7.78 | 6.43 | 0.20 |
| | 7.6 | TNT | 200×200 | 1.6 | 237 | 0.178 | 6.76 | 6.47 | 0.04 |
| | 8.2 | TNT | 200×200 | 1.6 | 237 | 0.192 | 7.13 | 7.37 | −0.03 |
| | 7.45 | TNT | 200 × 200 | 1.6 | 237 | 0.174 | 6.67 | 6.62 | 0.01 |
| | 12 | TNT | 200 × 200 | 1.6 | 237 | 0.281 | 10.55 | 13.68 | −0.22 |
| | 13.6 | TNT | 200 × 200 | 1.6 | 237 | 0.319 | 12.52 | 13.18 | −0.05 |
| 5 [30,31] | 5 | PEK I | 300 × 250 | 2 | 400 | 0.039 | 5.10 | 6 | -0.14 |
| | 50 | PEK I | 300 × 250 | 2 | 400 | 0.40 | 29.46 | 29.5 | −0.001 |
| | 70 | PEK I | 300 × 250 | 2 | 400 | 0.57 | 32.43 | 36 | −0.10 |
| | 50 | PEK I | 300 × 250 | 2 | 250 | 0.51 | 28.02 | 33.75 | −0.20 |
| | 60 | PEK I | 300 × 250 | 2 | 250 | 0.61 | 29.98 | 36.05 | −0.20 |
| | 70 | PEK I | 300 × 250 | 2 | 250 | 0.72 | 31.70 | 37.95 | −0.19 |

In these experiments, the experimental conditions, the chamber structure size and material, the box welding or fixing method, the type of explosives, and the height of the burst are different. The errors of the dimensionless deflection calculated with this analysis method are within 26%, which indicates that the method is effective. The error may exist because it does not take into account the height of the burst and box welding method, and these conditions have a great impact on the degree of deformation of the wall plate. In addition, as shown in Experiment 1, the scaling law of "imperfect similarity" [6] will also have an impact on the wall deformation results.

## 5. Conclusions

In this study, the damage mechanism of the box structure subjected to the internal blast load is analyzed based on the characteristics of the internal blast loadings and the

mechanism of the structural response. Different damage modes are classified according to the characteristics, and a dimensionless number and analysis method with clear physical meaning that considers structural deformation, inertia effect, and structural scale effect is proposed. Based on the study of isotropic box structures, a comparative study of damage effects of anisotropic box structures with different structural dimensions is carried out. The influence of the coupling relationship between the structural dimensions and the blast load on the damage modes is analyzed. Three competing mechanisms of material damage are also investigated, and the modes of breakage extension are analyzed and classified. Finally, the validity of the analysis method was verified by analyzing a large number of experiments. The main conclusions are as follows.

(1) The damage mechanisms and damage characteristics of the box structures subjected to different TNT-equivalent implosion loads were analyzed. The damage modes are divided into large deflection plastic deformation, central breach, corner or edge tensile tearing, and shear damage. The non-dimensional $D_{in}^*$ for the damage analysis applicable to internal explosions is proposed, and the equations describing the deformation of wall plates are presented. The variation of deflection and the open area with the non-dimensional $D_{in}^*$ for rectangular slabs of different structural dimensions under different implosion loads is further investigated. The variation of deflection and the open area of the wall plates subjected to different implosion loads with the non-dimensional $D_{in}^*$ is further investigated.

(2) Based on the study of isotropic box structures, a comparative study of the damage mechanism and damage characteristics of anisotropic box structures is carried out, and the effects of plate size and aspect ratio on the critical value at damage mode transition are analyzed in detail. For the same area of plates subjected to the same internal blast load, the smaller the aspect ratio $\gamma$ is (closer to a square plate), the larger the central deformation of the plate is. When the deformation exceeds the fracture strain, the damage mode of the wall plates changes from mode I to mode II. Subjected to the same internal blast load, wall plates with a smaller area are more likely to obtain the critical velocity and are more easily converted from damage mode III to IV. The variation of the central deflection of plates with different areas and aspect ratios is summarized to provide a basis in addition to providing support for predicting and assessing the damage level of anisotropic box structures.

(3) The opening extension mode of the wall was studied and classified with reference to the crack classification method. According to the basic logic of the damage model, the opening initiation of cracks belonged to slip-open (II) type openings. Distinguishing the different crack locations, the expansion direction process belongs to the tearing type expansion.

(4) The non-dimensional $D_{in}^*$ that can classify the damage mode, as well as the equations describing the plates' deformation that are applicable to the implosion damage analysis, are proposed. A large number of experiments are analyzed to verify the analysis method. In these experiments, the experimental conditions, box structural dimensions and materials, box welding or fixing methods, explosive types, and blast heights all vary. The dimensionless deflection errors calculated with this analysis method are within 26%, which indicates that the method is valid.

The dimensional analysis is limited to simple geometric configurations and cannot cope with complex structures. However, compared to the commonly used finite element software LS-DYNA, the time required for calculation is long and the threshold for use is high. The analysis method proposed in this paper indicates the approximate damage of the steel box structure in a very short time period, which makes it more suitable for emergency evacuation and rescue operations after an explosion, or for the rapid assessment of damage effects after a strike. The research results also lay the foundation for establishing the damage model of the box structure subjected to the internal explosion, which provides a reference and guide for engineers to design box structures and evaluate the damage degree of the structure under internal explosion.

**Author Contributions:** Conceptualization, L.-M.L.; Formal analysis, L.-M.L.; Funding acquisition, D.Z. and S.-J.Y.; Methodology, L.-M.L., D.Z. and S.-J.Y.; Project administration, D.Z.; Writing—original draft, L.-M.L.; Writing—review & editing, D.Z. and S.-J.Y. All authors have read and agreed to the published version of the manuscript.

**Funding:** The research reported in the present paper is supported by the National Natural Science Foundation of China (Projects No. 11902369, 11972371) and the Natural Science Foundation of Hunan province (Project No. 2021JJ30786). The research presented here is supported by the National Natural Science Foundation of China (Project No. 11972371&11902369) which is gratefully acknowledged.

**Conflicts of Interest:** The authors declare no conflict of interest.

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
