# Peer review of "Damage Mode Analysis of Steel Box Structures Subjected to Internal Blast Loading"

_applsci, doi:10.3390/app122110974_

Round 1

Reviewer 1 Report

Manuscript Applsci-1991107 reports on an investigation of the damages to steel boxes resulting from the air blast caused by the detonation of a high explosive charge, TNT. The authors observe a good agreement between results from dimensional analysis and numerical simulations obtained with the finite element software LS-DYNA.

The paper is well-written, and the results are discussed satisfactorily. I recommend the publication subject to minor revision. The authors should revise Section 2 to reflect better the usual presentation of the Vaschy-Buckingham (or PI) theorem on which their analysis is based. In particular, the numbers of dependent (n) and independent variables, i.e. with nondependent units, (p) should be explicit, and these variables fetched in a Table. They also should develop their discussion to emphasize that dimensional analysis is restricted to simple geometric configurations and cannot cope with complex structures in contrast to LS-DYNA which is widely used in industry for this type of evaluation.

Ref.[6], [11],[14],[16],[22],[29] incomplete, likely unaccessible
Ref. [19] vol#?, pages #?
Ref. [20] : write Livermore (not Liver-more)

Author Response

Response to Reviewer 1 Comments

Many thanks to the editorial board and reviewers for their constructive guidance, and the authors have made comprehensive and systematic revisions in the light of the review comments, as follows.

Point 1: The authors should revise Section 2 to reflect better the usual presentation of the Vaschy-Buckingham (or PI) theorem on which their analysis is based. In particular, the numbers of dependent (n) and independent variables, i.e. with nondependent units, (p) should be explicit, and these variables fetched in a Table.

Response 1: As suggested by the reviewer, a more detailed description of the use of the Vaschy-Buckingham (or PI) theorem has been provided in Section 2. The number of dependent variable (n) and independent variables, i.e. with independent variables (p), have been presented in Table 2.

Point 2: They also should develop their discussion to emphasize that dimensional analysis is restricted to simple geometric configurations and cannot cope with complex structures in contrast to LS-DYNA which is widely used in industry for this type of evaluation.

Point 3: Ref.[6], [11],[14],[16],[22],[29] incomplete, likely unaccessibl

Ref. [19] vol#?, pages #?  Ref. [20] : write Livermore (not Liver-more)

Response 3: As suggested by the reviewer, references have been revised and refined

Reviewer 2 Report

Referee report for “Damage mode analysis of steel box structures subjected to internal blast loading”. This paper is very well written with high-quality figures. Its topic and presentation make it well-suited for the journal. While this is not my area of expertise, I have no major comments and all the math that I looked at seemed to be correct. I only have minor comments to help clarify a few things in the paper and they are listed below.

    • What is γ (gamma)? I see it is defined as the aspect ratio in the conclusion but not before. A search for the term in the paper did not make it clear so the authors should update this so it is easy for future readers. I would suggest adding some wording for it to the table headers as well.

    • There should be a space between numbers and units. Sometimes you have it sometimes you do not.

    • Figure 5, the a, b, c, and d annotations are not clear. I can’t find them discussed in the paragraph that introduces this figure. I suggest you add text describing what they are to the captions.

    • Figure 5, what does the extra silver arrow represent? That one is confusing to me.

    • Figure 8, It should be “types of openings”

    • Figure 9, it would be helpful to point out what figure the markers are from. I see they are from 8, but this extra clarity would be nice.

    • Line numbers stop at 155?

    • line 63, missing a space between sentences.

    • Line 73 missing space between pi and theorem.

    • Line 91 missing space

    • missing space on line 104.

Author Response

Response to Reviewer 2 Comments

Many thanks to the editorial board and reviewers for their constructive guidance, and the authors have made comprehensive and systematic revisions in the light of the review comments, as follows.

Point 1: What is γ (gamma)? I see it is defined as the aspect ratio in the conclusion but not before. A search for the term in the paper did not make it clear so the authors should update this so it is easy for future readers. I would suggest adding some wording for it to the table headers as well.

Response 1: As suggested by the reviewer, the aspect ratio γ (gamma) was defined Section 3.4

Point 2: There should be a space between numbers and units. Sometimes you have it sometimes you do not.

Response 2: As suggested by the reviewer, we have checked and revised format

Point 3: Figure 5, the a, b, c, and d annotations are not clear. I can’t find them discussed in the paragraph that introduces this figure. I suggest you add text describing what they are to the captions.

Response: As suggested by the reviewer, we have add notes to a, b, c and d in Section 4.2

Point 4: Figure 5, what does the extra silver arrow represent? That one is confusing to me.

Response: The extra arrow is the direction of the profile and has been removed in accordance with expert opinion

Point 5: Figure 8, It should be “types of openings”

Response: As suggested by the reviewer, we have modified the image title

Point 6: Figure 9, it would be helpful to point out what figure the markers are from. I see they are from 8, but this extra clarity would be nice.

Response: As suggested by the reviewer, We have perfected the picture

Point 7:. Line numbers stop at 155? line 63, missing a space between sentences. • Line 73 missing space between pi and theorem.• Line 91 missing space • missing space on line 104.

Response 7: As suggested by the reviewer, We have checked and perfected the details
